# COMPARING SEMANTIC AND MORPHOLOGICAL ANALOGY COMPLETION IN WORD EMBEDDINGS

## ABSTRACT

Word embeddings have prompted great excitement in the NLP community due to their capacity for generalization to unforeseen tasks, including semantic analogy completion. Features such as color and category relationships have been examined by previous work, but this is the first research considering the morphological relationships encoded in word embeddings. We construct several natural experiments examining analogy completion across word stems modified by affixes, and find no evidence that Word2Vec, glove, and fasttext models encode these morphological relationships. We note that a special case of this problem is part-of-speech transformation, and note that the lack of support for part-of-speech analogies is surprising in the context of other successful cases of semantic inference using word embeddings.

## 1 INTRODUCTION

Prior work (Fejzo et al., 2018) has shown that compounding morphological learning is key to language acquisition and literacy development in children. When we learn a new language, we are able to quickly memorize known words and derive unknown words through our knowledge of root words and morphological features. This ability enables language learners to efficiently and accurately comprehend the meanings of enormous words, but can computers do the same?

This introduction reviews prevailing approaches to modeling semantic relationships between words, as well as some prior work on morphological and analogical analysis. Then, methods for quantifying the ability for word embeddings to represent semantic morphological features are proposed in the form of analogy-completion tasks. Experiments are run on three common word embeddings and results are summarized.

### 1.1 PRIOR WORK

Early popular methods for Natural Language Processing (NLP) relied on simplistic models trained over large datasets, such as the N-gram for statistical language modeling. However, although such models outperform many more sophisticated counterparts, a limitation is the size of obtainable data for training, which impedes accurate representations over a large corpus of words. As the corpus size increases, the computational resources required to handle the learning with this simplistic architecture also accumulate.

#### 1.1.1 WORD EMBEDDING MODELS

In 2013, a group led by Google scientist Tomas Mikolov (Mikolov et al., 2013a) proposed a novel continuous vector representation method along with two neural network models built for the task: Continuous Bag-of-Words (CBOW) Model and Continuous Skip-gram Model. Mikolov's model considered not only the proximity but also the similarity between related words in the vector space, and a multitude of degrees of linearity is encoded into the word embedding. With this property, using simple algebraic operations of the vector can formulate computer generate analogies comparable to those of humans. Pairwise semantic relationships within a subset of words can also be represented using the linear properties of the embedding space.

In a subsequent publication, Mikolov (Mikolov et al., 2013b) proposed several optimization methods, such as subsampling the Skip-gram Model. To encode idiomatic information, simple phrases composed of individual words were also trained as separate entries. The optimized model could train on 100 billion words a day, with a total data size of billions compared to a few millions of words in other models.

Word embedding has become a central topic in the field of Computational Linguistics and Natural Language Processing. The Word2vec model (Mikolov et al., 2013a), utilizes Continuous Bag-of-words (CBOW), which calculates nearby context words but disregards any additional syntactical information. Due to its simplifying nature, CBOW tends to rely on multiplicity rather than syntax. (Mikolov et al., 2013a)

GloVe (Pennington et al., 2014) is another word embedding model, first published by Jeffry Pennington and his team from Stanford University in 2014. Like Word2Vec, GloVe utilizes a similar unsupervised process, but the difference is that it combines local word context information and global word co-occurrence statistics.

In contrast to Word2Vec and GloVe, FastText (Joulin et al., 2016) by Facebook AI Research (FAIR) is based on a supervised sentence classification task. The task structure encourages word embeddings that can be added together to derive sentence-level representation. It can train on large datasets relatively quickly and run with less computational consumption than prior models.

### 1.1.2 CONTEXT-SENSITIVE SEQUENTIAL MODELS

Some more recent approaches to general NLP tasks involve context-sensitive word-embeddings using the Transformer Model, first introduced in 2017 Vaswani et al. (2017). Similar to Recurrent Neural Networks (RNN), Transformer Models are designed to process sequential input, but the key difference is that it utilizes the "Attention Mechanism", which weights the context vectors according to their estimated importance and forms connections within a sentence. This method enhances encoding of long-term dependencies and improves the overall performance of the model. Due to the high efficiency of the Transformer model, several highly general pre-trained models have been designed, such as BERTDevlin et al. (2018), GPT-3Brown et al. (2020), and ELMOPeters et al. (2018), for a variety of tasks with fine-tuning. These context-sensitive embeddings are more expressive than the static word-embedding models that map each word (or idiomatic phrase) to a single vector, regardless of context, but they are much harder to study semantically as any metrics for word similarity become necessarily contextual.

### 1.1.3 SEMANTIC EMBEDDINGS FROM MORPHOLOGICAL FEATURES

Several approaches have been proposed to specifically encode morphological features into word embedding models. One method is character-based learning on word segmentsCao & Rei (2016), which divides words into segments, treating morphemes separately. However, the character-based model performed worse on semantic similarity and semantic analogy tasks than traditional word-based models.

Another is vector representation by composing characters using bidirectional LSTMs(Ling et al., 2015). This model excels in morphologically rich languages, languages that indicate part of speech by adding affixes rather than changing the position in a sentence, but learning complicated rules for how characters link together is inefficient on larger corpora.

### 1.1.4 WORD EMBEDDING ANALOGIES

There is also some prior work examining vector-based analogies in word embedding space. For example, one research paper (Bolukbasi et al., 2016) examines the semantic quality of gender as a difference vector in embedding space for the w2vNEWS embedding (a variant of Word2Vec). The paper focuses specifically on gender bias in word representation and use, and uses similar methods to this paper to explore the semantic representation of gender.

## 1.2 CONTRIBUTIONS

The contributions of this work are two-fold. First, systematic experiments around morphological analogy completion reveal that Word2Vec, gloVe, and FastText word embeddings all fail to capture the same quality of semantic information around meaning-modifying prefixes (such as un-, re-, or anti-) as between oft-touted "common-sense" examples. Furthermore, these models fail to capture the relationships encoded morphologically by part-of-speech modifying suffixes (such as -tion, -ing, or -ly). Second, a broad statistical examination of word-pair distance-vectors corroborates the absence of morphological features in these commonly-used word-embedding models.

## 2 METHODS

This work consists primarily of exploratory experiments on pre-trained language models. This section describes the models, datasets, and analogy-completion tasks used for this work, as well as the methods of statistical analysis used to examine the semantic representation of morphological features.

## 2.1 PRETRAINED MODELS

This work uses three popular pre-trained word-embedding models: Word2Vec (Mikolov et al., 2013a), GloVe (Pennington et al., 2014), and FastText (Bojanowski et al., 2016). Each of these pre-trained models represent words (and some idiomatic phrases) as a single vector. This work utilized pre-trained variants with 300-length vectors for each model. These word-embedding models are static, or context-insensitive, in that the same word in different sentences always receives the same embedding. This means that words with multiple highly distinct meanings (polysemy) are sometimes ill-represented, but has the advantage that the relationship between two words across all contexts can be modeled as the difference between those two vectors.

## 2.2 ANALOGY COMPLETION

Word embeddings achieve analogy completion with algebraic vector operations. For example, in the classic case of "man is to king as woman is to queen", we can obtain the word "queen" by adding the vector of "woman" to "king" and subtracting the vector of "man". This operation allows for pairwise analogy for any given three words, by searching for the word whose embedding is nearest to the predicted vector. The capacity for word embeddings to solve these kinds of analogy-completion problems (without explicitly training for them) has been a compelling demonstration of the depth and versatility of the word embedding models. As demonstrated in table 2.2 using 10 hand-crafted examples, Word2Vec (by way of example) appears to accurately represent a variety of semantic features. It is noteworthy that of the 10 examples, 8 are perfectly completed, 1 has a plausible but unintended answer, and only 1 has an unreasonable answer.

Table 1: Examples of Analogy Completion with Semantic Information using Word2Vec

| Parent Analogy | Target Word | Computed Answer | Success |
|---|---|---|---|
| man : king as woman : queen | queen | queen | Yes |
| student : learn as teacher : teach | teach | teach | Yes |
| big : small as wide : narrow | narrow | narrow | Yes |
| man : woman as uncle : aunt | aunt | aunt | Yes |
| little : big as dwarf : giant | giant | dwarfs | No |
| boy : girl as man : woman | woman | woman | Yes |
| high : low as up : down | down | down | Yes |
| paris : france as tokyo : japan | japan | japan | Yes |
| brother : sister as grandson : granddaughter | granddaughter | granddaughter | Yes |
| food : eat as toy : play | play | Legos | No |

## 2.3 Most-Common-Word Datasets

The experiments in this paper used a public dataset of most common words from Google's Trillion Word Corpus [1], pre-processed by Google's N-gram models. This dataset contains the first ten thousand most common words calculated from 1,024,908,267,299 words of text analyzed by Google's AI research teams. To examine morphological feature representation in embedding space, a list of the 22 most common prefixes and 30 suffixes [2] was also used.

A dataset of word pairs consisting of a root word and an affix-root compound was generated for each affix. The most common words were initially filtered for the presence of the affix morpheme, and then were further filtered to those with valid roots when truncating the affix. Note that some valid pairs were omitted by the procedure because the affix changed the spelling of the root component (e.g. happy : happily), and some pairs that were generated were semantically implausible (e.g. member : remember). In total 6730 pairs of words (affix-modified and root words) were generated. This will be called the "affix-pairs dataset".

## 2.4 Morphological (Affix) Analogy Completion

A large number of pairwise morphological analogy completion tasks were generated by selecting two words with the same affix and their corresponding root words (say, a : b and c : d), then applying algebraic vector operations (b - a + c) and comparing the resulting vector to the closest words in the embedding space.

Grouped morphological analogy completion tasks were also created by taking the arithmetic mean of the difference vectors for each affix, and then testing whether each modified word could be predicted by added the mean difference vector to the root word.

## 2.5 Distributions of Distance Vectors

To further explore morphological feature encoding in word embeddings, the distribution of differences (and differences of differences) of word-pair vectors is analyzed. In particular, this distribution is compared to the distribution of differences between random pairs of words selected uniformly from the 10000 most common words. If morphological features are well represented by the embedding space, the norms of the difference vectors (between the root word and the modified word) should be tightly clustered (more tightly than the differences between random words). Furthermore, the difference *between the word-pair difference vectors* (the differences between "modification vectors") should be close to zero. (In other words, if the same kind of semantic analogy exists for morphological features as for other semantic features, the difference vectors between root words and the words modified by the same affix should be very similar.)

## 3 Experimental Results

This section considers, in-turn, analogy-completion tasks, distributions of difference vectors, and structural similarities in the morphological relationships encoded by each of the word embeddings.

## 3.1 Do Word Embeddings Solve Pairwise Morphological Analogies?

No. **Across word pairs using the same affix, no tested pair of word-pairs resulted in successful analogy completion in Word2Vec, GloVe, or FastText.** Pairs of word-pairs were randomly sampled for each affix and embedding, and word-pairs with the most similar difference vectors (as measured by Euclidean norm) were also explicitly tested. The 10 pairs of word pairs (from the affix-pairs dataset) with most similar difference vectors in Word2Vec are summarized in Table 3.1. This result stands in stark contrast to the non-morphological semantic analogies, where many previously untested analogies are completed successfully.

---

[1] https://ai.googleblog.com/2006/08/all-our-n-gram-are-belong-to-you.html
[2] https://www.scholastic.com/content/dam/teachers/lesson-plans/migrated-files-in-body/prefixe

Table 2: Examples of Analogy Completion with Morphological Information

| Parent Analogy | Target Word | Computed Answer | Success |
|---|---|---|---|
| fixed : fix as fined : fine | fine | fining | No |
| presented : present as allowed : allow | allow | permitted | No |
| supported : support as signed : sign | sign | signing | No |
| informed : inform as worked : work | work | working | No |
| sorted : sort as covered : cover | cover | kind | No |
| intended : intend as viewed : view | view | see | No |
| extended : extend as owned : own | own | owns | No |
| played : play as liked : like | like | loved | No |
| returned : return as blamed : blame | blame | blames | No |
| passed : pass as ended : end | end | ending | No |

## 3.2  DO WORD EMBEDDINGS SOLVE AVERAGE MORPHOLOGICAL ANALOGIES?

No. **Across *all* affixes and the three considered embeddings, there was not a single instance of successful analogy completion using the average difference vector for a given affix.** This outcome generally agrees with the pairwise analogy result, but further demonstrates that the pairwise result is not merely a consequence of unbiased noise.

## 3.3  DO AFFIX-WORD-PAIR DIFFERENCES CLUSTER MORE TIGHTLY THAN RANDOM PAIRS?

Unclear. As seen in Figure 3.3, the difference vectors for "ed" (selected as an example due to its comparatively small average distribution of difference vectors) have a different distribution shape but have comparable range to the differences of random pairs of words. **No affix under consideration demonstrated a substantially narrower distribution.**

## 3.4  DO DIFFERENCES OF DIFFERENCE VECTORS CLUSTER NEAR ZERO?

No. As shown in Figure 3.3, differences between word-modification vectors for the suffix "-ed" have a mean well away from zero.

As shown in Figure 3.4, **the mean difference between modification vectors for pairs involving the same affix was well away from zero for all affixes considered and in many cases exceeded the difference between random words**.

## 3.5  DO DIFFERENT WORD EMBEDDINGS ENCODE MORPHOLOGICAL FEATURES STRUCTURALLY?

Yes. Although this result is surprising in contrast to the other results in this paper, the three word embeddings considered show a high correlation between the rank order of affixes by difference of modification vectors, as shown in Table 3.5. Examples such as "tion", "ious", and "de" have high mean norms compared to others such as "s", "ed", and "ty" for all models. This suggests that **while some affixes are better represented than others, these patterns are mostly consistent throughout all three models**. Note that GloVe and Word2Vec have the most similar structural trends, which is likely a result of their more similar training procedure and data (relative to FastText).

Although this kind of structure is not predictive in the sense of enabling analogy-completion, it does show that morphological features are reflected in the topology of the embedding space. For example, across all of the embeddings, pluralization is the most analogous operation, perhaps because words and their plurals have very similar meanings to begin with.

## 3.6  SUMMARY

These experiments demonstrate that morphological features with semantic content are generally not represented in commonly used word embeddings, at least not to the same extent that non-morphological semantic qualities, such as gender, age, location, or opposites seem to be encoded.

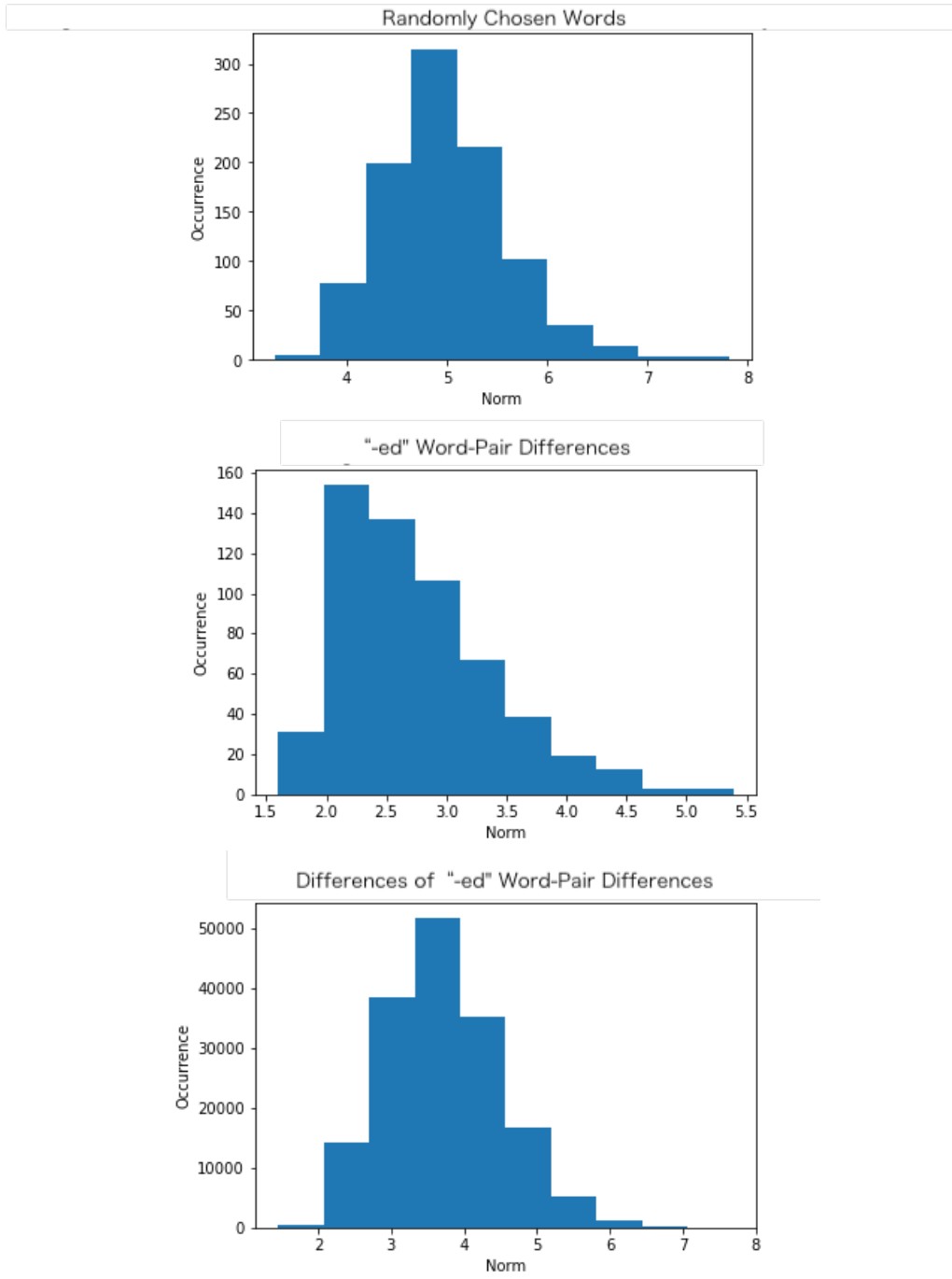

Figure 1: From top to bottom: the distribution of (Euclidean) norms of difference vectors between pairs of words chosen uniformly from the 10000 most common; the distribution of norms of difference vectors for root words and root words plus "ed"; the distribution of norms of differences *between pairs of difference vectors* for word-pairs from the "-ed" set. All numbers from Word2Vec.

At the same time, there appear to be structural similarities (relating to the spread of modification vectors) between embeddings produced by different methods. Returning to the motivating question of the paper, extant word embeddings do reflect some properties of morphological features, but do not generalize across morphological analogies as humans do.

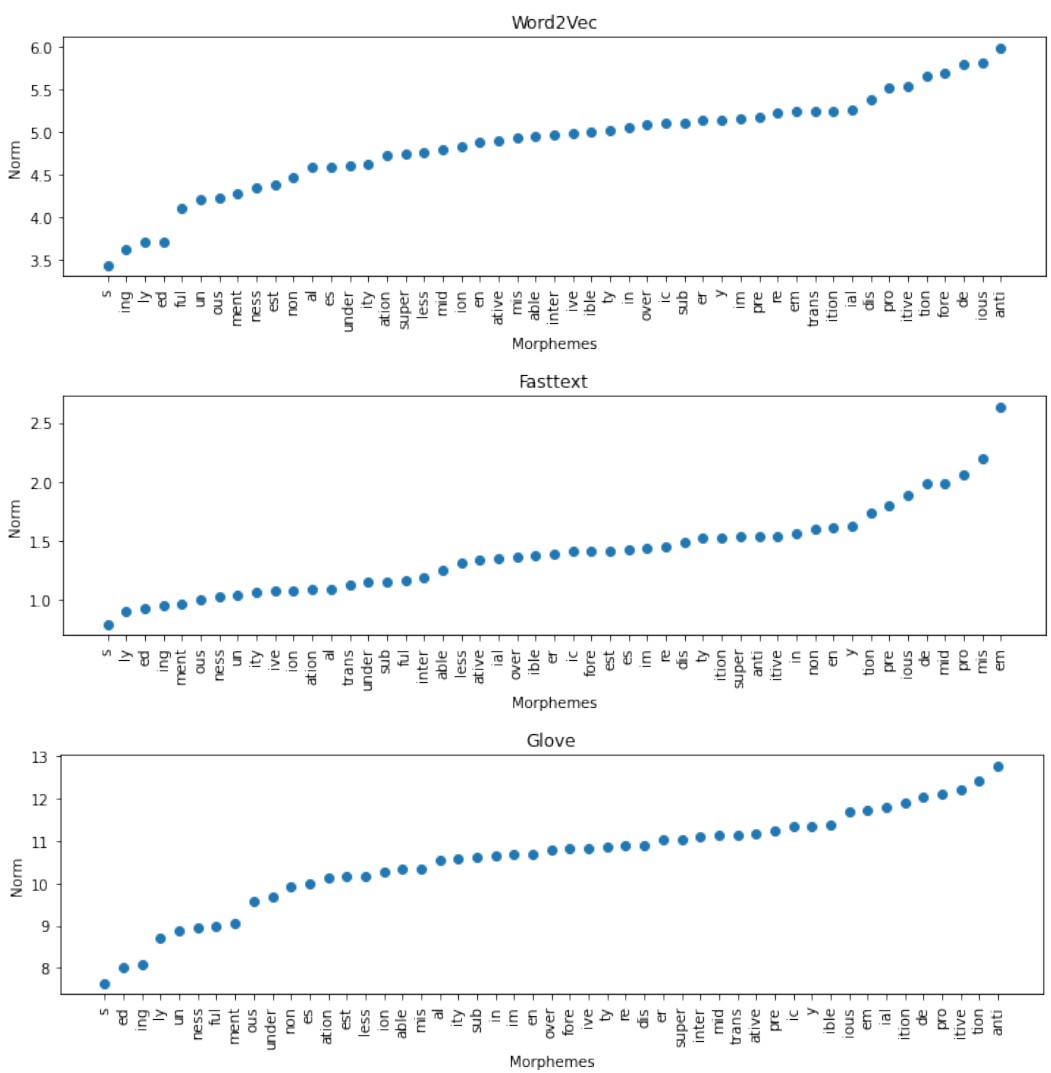

Figure 2: Mean Euclidean norm of differences between word-modification vectors for three word embeddings are plotted for each affix from the suffix and prefix list.

Table 3: Pairwise Spearman's Rank Coefficient for Word2Vec, FastText, and GloVe.

| Model Names | Coefficient |
|---|---|
| Word2Vec + FastText | 0.66449 |
| Word2Vec + GloVe | 0.88337 |
| FastText + GloVe | 0.58122 |

# 4 DISCUSSION

Here, the limitations and implications of the work are discussed. While the experiments conducted were confined to common English words, three static word-embedding models, and analogy completion experiments, they are very suggestive that morphological features may not be well-represented by fairly modern word-embedding approaches.

### 4.0.1 LIMITATIONS

One limitation of this work is the simplification of the morphological model considered. We excluded words with modified root when combined with an affix due to the complexity of such modification in English and the difficulty of its inclusion into the experiment. If we were able to encompass all words with affix regardless of transformation, we would acquire a larger and more representative morphological model, although this seems unlikely to alter the qualitative outcomes of these experiments.

The scope of our experiment is also limited to three commonly-used word-embedding models: Word2Vec, FastText, and GloVe. To gain a better understanding of word embedding as a whole, an examination of other models of different characteristics and publication times is likely to yield a more well-rounded result. Furthermore, all embedding models considered here are contextless, meaning that there is no sentence-level information to handle challenges such as polysemy. It is much more difficult to construct rigorously quantitative analogy-completion experiments for context-sensitive transformer-based models, but the authors observe anecdotally that naive initial attempts failed to elicit clear evidence of morphological feature encoding.

### 4.0.2 IMPLICATIONS

This work at once reproduces oft-touted instances of semantic analogy completion (for example, woman:queen::man:king) and yet also shows that many morphological relationships that reflect deep semantic features of human language do not have comparable representation in Word2Vec, FastText, and GloVe. Furthermore, statistical analysis of word-pair distance-vectors suggests that the words in question are not near to each other in embedding space, and their difference vectors do not share a common direction.

This is a surprising and unfortunate result. Without the ability to quantitatively represent the effect of morphological modification of words, it will be difficult to develop language models that reproduce humans' ability to generate and understand novel words, as well as to generalize from much smaller sets of textual examples.

In fact, this shortcoming may be a natural consequence of the Continuous-Bag-of-Words Representation. Since this approach to estimating word relationships based on frequency of proximity largely ignores the relationship of words that *do not* co-occur, and many morphologically-modified words are either semantically or grammatically incompatible (for example, because of verb-tense agreement), morphological feature relationships may be relatively unconstrained by the model training objective.

## 5 CONCLUSION

Our experiments show that morphological features are at best indirectly represented in common CBOW word embeddings. Both the empirical distributions of embedding-space distances and performance on analogy completion tasks suggest this conclusion.

The questions of *why* morphological features are under-represented remains unanswered. Is this just a pitfall of CBOW-based embeddings? How could we address this problem? Does this suggest, by contrast, a pattern in human acquisition of language?

Methods that explicitly utilize morphological learning might significantly improve the efficiency and accuracy of word embedding models and provide a better solution to many NLP problems. Such models could learn from a much smaller set of root words compared to Word2vec, which treats root words and its derivatives as individual entries. A reduction in the complexity of the model can decrease the computational resources required and increase the accuracy of the embedding. Mimicking natural language learning of human more closely, word embedding models that contain morphological learning could potentially lead to revolutionary performance in zero-shot NLP tasks, including those with novel words.

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
