# OpenReview forum: "Comparing semantic and morphological analogy completion in word embeddings"
_ICLR.cc/2023/Conference — Submitted to ICLR 2023_

### Official Review · Reviewer_REfa · 2022-10-24

**Confidence:** 4
**Correctness:** 2
**Technical Novelty And Significance:** 1
**Empirical Novelty And Significance:** 1
**Recommendation:** 1

**Clarity, Quality, Novelty And Reproducibility:**

Clarity - Ok. I can follow the main points without much effort. Phrasing in a few places needs refinement.

Quality - poor. Related work severely under-surveyed leading to unclear positioning and false claims. Important details missing about dataset and experimental settings, undermining the strength of the empirical evidence and the validity of the claims. Dubious claims regarding related works and implications.

Novelty - poor. Trivial or known findings.

Reproducibility - poor. Cannot reproduce with important details missing and no code or resources provided.


**Strength And Weaknesses:**

## Strengths
None

## Weaknesses
1. Related works are severely under-surveyed. Pos tagging and morphological segmentations (e.g. Ruokolainen et al., 2016) are two popular tasks in NLP. A number of works have examined the abilities of word embedding in capturing morphological information (e.g. Botha et al., ICML 2014; Cao & Rei, 2016; Cotterell & Schütze, TACL 2018; Jinman et al., Findings 2020), or POS-tagging (e.g. Ling et al., NAACL 2015; Pinter et al., EMNLP 2017; Zhao et al., EMNLP 2018). Just name a few. The authors mention or discuss none of the existing investigations. I suggest the authors read some of the surveys, e.g. Yang et al. (2018), recent evaluations of word embedding models (e.g. Wang et al., 2019; Ghannay et al., LREC 2016), or try search on Google Scholar using keywords like word2vec, word embedding, syntax, morphology, pos tagging.
2. The descriptions of previous works (word2vec, GloVe and FastText) are misleading. One main difference between word2vec and GloVe is that word2vec uses localized losses similar to stochastic gradient descent whereas GloVe uses matrix factorization. FastText uses a similar objective as word2vec but added on a bunch of tricks for performance and efficiency.
3. Important details about dataset missing, affecting the validity and reproducibility. In Sec 2.3, the list of chosen affixes is not provided. The criteria and the list of chosen roots is not described. It is not clear if the authors take careful steps beyond simple string matching to make sure if the selected "affix pairs" are true morphological conjugates, rather than mere look-alikes.
4. There exists carefully crafted datasets for word affixes, e.g. Lazaridou et al. (ACL 2013). The authors do not discuss or justify the need or benefit of their new dataset.
5. Important details about experiment setting missing, affecting the validity and reproducibility. In Section 3, the authors do not specify which versions of pre-trained w2v, glove, and fasttext are used. The authors only mention that they use the 300d version. However, each of the embedding methods has many publicly available checkpoints trained over different corpus, in different (combination of)  languages, and with different hyperparameters. Some of the hyperparameters have been shown to drastically change the downstream performance of the word vectors,  e.g subword size of fastText (Bojanowski et al., TACL 2017, Table 4), corpus domain (Lat et al., 2016), number of negative examples (Levy et al., 2014), window size. Window size has especially been shown to affect the syntax-semantics trade-off in learned embeddings.
6. Experiment support is weak for the claims. The authors heavily rely on anecdotal examples and study of a single case rather than quantitative analysis. In Table 1, 2, the authors use hand-picked examples to show that word embedding is able to complete semantic analogy but not syntactic analogy, where top-k accuracy or the error/correlation between the predicted and the correct word vector could be more convincing. In Fig 1, the authors use histogram to show the distribution of norms of word vectors related to "-ed", where numeric statistics (mean, variance, etc) of all cases could be more convincing.
7. Findings are trivial or known. The ability of word embeddings on word analogy (Sec 2.2 and Table 1) is known knowledge and has been reported in many previous works. Search works that use WordSim353 (Finkelstein et al., 2001), s RareWord (Luong et al., 2013), Card-660 (Pilehvar et al., 2018) and other similar word analogy benchmark. Sec 3.5, it is known that w2v vector norms correlated to word rank.
8. Dubious claims. See LPs regarding P8.

## Localized points (LPs)
1. Use \citet{} for "Smith et al. (2022)" style citation and \citep{} for  "(Smith et al., 2022)" style citation.
2. P4, Footnote 2. Link to the common prefixes not accessible.
3. P4, Sec 2.3. Do you manually inspect and exclude "implausible" instances? Finding true morphological affixes is much trickier than string matching. For example, "replacement" can be analyzed as replace/ment but not re/placement.
4. P5, Sec 3. Where is Figure 3.3, Figure 3.4 and Table 3.5? They probably mean Figure 1 (middle), Figure 1 (bottom), Figure 2. Please fix this. Maybe use \label and \ref?
5. P5, Sec 3.3. Could report variance for all affixes instead of just showing a plot for one affix.
6. P5, Sec 3.4. I guess the authors wanted to say the distribution of difference vector has a large variance. Computing the variance would be much clearer.
7. P8, "Furthermore, statistical analysis of word-pair distance-vectors suggests that the words in question are not near to each other in embedding space, and their difference vectors do not share a common direction." Why should they be near? The evidence in this paper (Figure 1 middle) indeed show that the vector for "[root]ed" is indeed closer to "[root]" than random word pairs.
8. P8, "Without the ability to quantitatively represent the effect of morphological modification of words, it will be difficult to develop language models that reproduce humans’ ability to generate and understand novel words, as well as to generalize from much smaller sets of textual examples." We have seen a plethora of evidence of the two.

## References
1. Two/Too Simple Adaptations of Word2Vec for Syntax Problems (Ling et al., NAACL 2015), https://aclanthology.org/N15-1142/
2. Compositional Morphology for Word Representations and Language Modelling. Jan Botha, Phil Blunsom Proceedings of the 31st International Conference on Machine Learning, PMLR 32(2):1899-1907, 2014. https://proceedings.mlr.press/v32/botha14.html
3. A Joint Model for Word Embedding and Word Morphology (Cao & Rei, 2016), https://aclanthology.org/W16-1603/
4. Ryan Cotterell, Hinrich Schütze; Joint Semantic Synthesis and Morphological Analysis of the Derived Word. Transactions of the Association for Computational Linguistics 2018; 6 33–48. doi: https://doi.org/10.1162/tacl_a_00003
5. PBoS: Probabilistic Bag-of-Subwords for Generalizing Word Embedding (Jinman et al., Findings 2020), https://aclanthology.org/2020.findings-emnlp.53/
6. Mimicking Word Embeddings using Subword RNNs (Pinter et al., EMNLP 2017), https://aclanthology.org/D17-1010/
7. Generalizing Word Embeddings using Bag of Subwords (Zhao et al., EMNLP 2018), https://aclanthology.org/D18-1059/
8. Li, Yang, and Tao Yang. "Word embedding for understanding natural language: a survey." Guide to big data applications. Springer, Cham, 2018. 83-104. https://link.springer.com/chapter/10.1007/978-3-319-53817-4_4
9. Teemu Ruokolainen, Oskar Kohonen, Kairit Sirts, Stig-Arne Grönroos, Mikko Kurimo, Sami Virpioja; A Comparative Study of Minimally Supervised Morphological Segmentation. Computational Linguistics 2016; 42 (1): 91–120. doi: https://doi.org/10.1162/COLI_a_00243
10. Wang, Bin, et al. "Evaluating word embedding models: methods and experimental results." APSIPA transactions on signal and information processing 8 (2019). https://www.cambridge.org/core/journals/apsipa-transactions-on-signal-and-information-processing/article/evaluating-word-embedding-models-methods-and-experimental-results/EDF43F837150B94E71DBB36B28B85E79
11. Word Embedding Evaluation and Combination (Ghannay et al., LREC 2016), https://aclanthology.org/L16-1046/
12. Compositional-ly Derived Representations of Morphologically Complex Words in Distributional Semantics (Lazaridou et al., ACL 2013), https://aclanthology.org/P13-1149/
13. Piotr Bojanowski, Edouard Grave, Armand Joulin, Tomas Mikolov; Enriching Word Vectors with Subword Information. Transactions of the Association for Computational Linguistics 2017; 5 135–146. doi: https://doi.org/10.1162/tacl_a_00051
14. Lai, Siwei, et al. "How to generate a good word embedding." IEEE Intelligent Systems 31.6 (2016): 5-14. https://ieeexplore.ieee.org/abstract/document/7478417, https://arxiv.org/abs/1507.05523
15. Levy, Omer, and Yoav Goldberg. "Neural word embedding as implicit matrix factorization." Advances in neural information processing systems 27 (2014).  https://proceedings.neurips.cc/paper/2014/hash/feab05aa91085b7a8012516bc3533958-Abstract.html


**Summary Of The Paper:**

The authors aim to study the ability of pre-trained word embeddings in representing morphological relations between words. The authors constructed a dataset of words with affixes and their roots. The authors conduct experiments on word analogy tasks concerning word affixes and explore the distribution of the difference of the word vectors of word pairs.

**Summary Of The Review:**

The paper requires significant improvements in all key aspects to meet the bar for publication. I suggest that the authors start with a thorough survey and review of existing work and then perhaps rethink the approach and the story.

---

### Official Review · Reviewer_L7qJ · 2022-10-24

**Confidence:** 4
**Correctness:** 3
**Technical Novelty And Significance:** 1
**Empirical Novelty And Significance:** 1
**Recommendation:** 3

**Clarity, Quality, Novelty And Reproducibility:**

While the goal of this paper is quite clear, its novelty is questionable. While the paper presents extensive studies of the morphological (affix) information in a word embedding. There are related several works that the authors omitted. For example,
- [Improve word embedding using both writing and pronunciation](https://journals.plos.org/plosone/article?id=10.1371/journal.pone.0208785)
- [Unsupervised Morphology Induction Using Word Embeddings](https://aclanthology.org/N15-1186.pdf)
- [Morphological Word-Embeddings](https://aclanthology.org/N15-1140.pdf)

In terms of the experiments, it is unclear how the authors filter word pairs. While the word embeddings studied in the paper are publicly available, the list of word pairs was not provided to reproduce the results.

**Strength And Weaknesses:**

Strengths:
1. The paper presented a series of investigations and reveal interesting insights about word embeddings.

Weaknesses:
1. Since most of the NLP methods have moved beyond standalone word embeddings, this makes the insight revealed in this paper not as relevant. The authors could try to conduct a similar study on contextual embedding approaches.
2. What is being captured in word embeddings could depend on the context size (shorter context, more syntactic), this is an essential aspect to include in the study.
3. The key insight from the study is not quite novel. There were several works augmenting morphological information to the word embeddings (https://aclanthology.org/N15-1140.pdf or https://arxiv.org/abs/1407.1687). Perhaps, the authors could try to study these methods.

**Summary Of The Paper:**

This paper studied the morphological relationships encoded in word embeddings learned from traditional approaches such as Word2Vec, GloVe, and FastText. The authors used the analogical reasoning task as an instrument to investigate morphological information. The experiments conducted on 6,730 pairs of words (-affix) revealed that the word embeddings failed the morphological analogy task. A series of morphological investigations showed that there was no consistent trend.

**Summary Of The Review:**

I do not recommend the paper to ICLR 2023 because the paper lacks a significant literature review and the impact of the paper is not relevant to the current NLP research.

---

### Official Review · Reviewer_uwiD · 2022-10-24

**Confidence:** 3
**Correctness:** 1
**Technical Novelty And Significance:** 1
**Empirical Novelty And Significance:** 1
**Recommendation:** 1

**Clarity, Quality, Novelty And Reproducibility:**

The methodological novelty and technical rigor of the paper is not sufficient to qualify as an ICLR submission. Also there are no useful take-home messages that can benefit those who are still using static embeddings. Given the empirical evaluation, I am not ready to accept the claims made in the paper.

**Strength And Weaknesses:**

Weaknesses

While it is interesting to note that analogy completion does not function the same way for morphology in static models, I think it is still too strong to claim that these models do not encode morphological relationships. Previous work has shown that representation from static embedding such as word2vec, GloVe etc. capture morphology, by training classifiers from the feature vectors towards the task of morphological tagging. A lot of work on probing linguistic properties for static and contextualized embedding have shown that the models capture word morphology during earlier epochs of training. For example see

Understanding the learning dynamics of word2vec

Another recent work is from Musil et al (2019) where they use unsupervised clustering of word2vec vectors to capture derivational relationships.

I could not understand why the methodology can’t be applied to contextualized embedding? Studying deep transformer pLMs would be much more interesting where previous research has shown that lower layers learn morphological features whereas the higher and upper layers capture semantic properties using probes. The earlier layers predominantly reconstruct word structure which is disintegrated from subword segmentation. I am hopeful that applying the method in the earlier layers of pLMs would reveal interesting findings.




**Summary Of The Paper:**

The semantic understanding of word representation is often established through the analogy completion task where vector arithmetics such as King - man + woman = Queen example is notoriously used to establish the static embedding models like Word2Vec learn semantic relationships. Learning word morphology is fundamental to learning a language and the authors conjecture that the static embedding models don’t capture this knowledge by formulating analogy completion across word stems modified by affixes. The examples like sorted: sort as covered: cover fail. The authors also analyzed the distribution of distance vectors to see if morphological features are well represented in the embedding space.


**Summary Of The Review:**

The authors used an off-the-shelf method to question the existence of morphological knowledge in the static embedding models. Their approach is very limited for the claim made in the paper, especially when previous research has shown through other methods that these models do learn word morphology. The paper does not have methodical novelty, and the arguments are weak. Also the paper does not experiment with or discuss state-of-the-art to draw any comparative analysis. It is not clear how the community can benefit from this paper in any sense.

---

### Official Review · Reviewer_CtMu · 2022-10-24

**Confidence:** 4
**Correctness:** 3
**Technical Novelty And Significance:** 2
**Empirical Novelty And Significance:** 2
**Recommendation:** 3

**Clarity, Quality, Novelty And Reproducibility:**

The clarity of the paper is good.

I am concerned with a little novelty as the paper adds a little to the current knowledge about word2vec like models.

**Strength And Weaknesses:**

Strengths:

- Paper represents a clear useful nugget of research answering a few research questions. It allows for better understanding of word2vec capabilities. These may be useful for building some morphological analysers.

Weaknesses:

- Little novelty. Paper clearly adds some knowledge about word embeddings, but I would expect a more large scale study from a full paper.

- The static word embeddings we gaining their momentum about 10 years ago and since then Transformers and other deep pre-trained language models addressed the problem of morphology in a different way i.e. through BPE encoding. Authors do not mention or put in context of these more modern development their research. Maybe its a good way to improve the paper though larding the scope and adding more models to consider (ask questions how they deal with morphology).  One opportunity is to to extract contextualised word embedding from these models and add to comparison.

**Summary Of The Paper:**

The paper aims at revisiting study on analogy capabilities of word2vec models. More specifically, morphological task is studied with Glove, fasText and word2vec (CBOW) word embeddings. Contrary to examples suggested in the original word2vec paper, word2vec model performs poorly on a line of morphological tasks.

**Summary Of The Review:**

Reasonable questions to address, but (i) has a limited novelty, and (ii) do not put work in context of more modern approaches such as Transformers with BPE approach to morphology.

---

### Decision · Program_Chairs · 2023-01-20

**Decision:**

Reject

**Justification For Why Not Higher Score:**

Lack of novelty, not a problem of high importance

**Justification For Why Not Lower Score:**

no lower score possible

**Metareview: Summary, Strengths And Weaknesses:**


SUMMARY

The paper revisits the analogy capabilities of word2vec-type
models. More specifically, morphological tasks are studied
with Glove, fasText and word2vec (CBOW) word
embeddings. Contrary to what is suggested in the original
word2vec paper, word2vec performs poorly for particular
types of morphological tasks.

STRENGTHS

The paper represents a clear useful nugget of research.

The paper contributes to a better understanding of word2vec
capabilities.

The results may be useful for building morphological
analyzers.

WEAKNESSES

Lack of novelty (both in methods and findings)

Lack of discussion and comparison with related work

Static word embeddings are no longer seen as a topic of
prime importance. Contextualized embeddings are not
addressed.

The work can be seen as a type of probing. Only one probing
method is investigated. It is not justified to conclude that a
particular type of information is not present just because
one probing method fails.

Experiments not described in sufficient detail